# Identification, Association of Natural Variation and Expression Analysis of *ZmNAC9* Gene Response to Low Phosphorus in Maize Seedling Stage

**DOI:** 10.3390/plants9111447

**Published:** 2020-10-27

**Authors:** Javed Hussain Sahito, Xiao Zhang, Haixu Zhong, Xuan He, Chen Zhen, Peng Ma, Bowen Luo, Dan Liu, Ling Wu, Muhammad Hayder Bin Khalid, Hakim Ali Sahito, Zeeshan Ghulam Nabi Gishkori, Asif Ali, Shibin Gao

**Affiliations:** 1Key Laboratory of Crop Gene and Exploration and Utilization in Southwest Region, Maize Research Institute, Sichuan Agriculture University, Chengdu 611130, China; javedhussainsahito@yahoo.com (J.H.S.); hunterzap@163.com (X.Z.); zhonghaixuccc@163.com (H.Z); hexuan1312@163.com (X.H.);cz765826100@126.com (C.Z.); mapeng561@163.com (P.M.); bowenluo_sicau@163.com (B.L.); liu2883508@126.com (D.L.); wtl.570@163.com (L.W.); 2Maize Research Institute, Sichuan Agriculture University, Chengdu 611130, China; haider2323@gmail.com; 3Faculty of Natural Sciences, Shah Abdul Latif University, Khairpur 66020, Sindh, Pakistan; hakim.sahito@salu.edu.pk; 4Rice Research Institute, Sichuan Agricultural University, Chengdu 611130, China; 2018611002@stu.sicau.edu.cn (Z.G.N.G.); asifali@sicau.edu.cn (A.A.)

**Keywords:** maize, identification, *ZmNAC9*, association, expression analysis, phosphorus deficiency

## Abstract

Phosphorus (P) is an essential macroelement supporting maize productivity and low-P stress is a limiting factor of maize growth and yield. Improving maize plant tolerance to low P through molecular breeding is an effective alternative to increase crop productivity. In this study, a total of 111 diverse maize inbred lines were used to identify the favorable alleles and nucleotide diversity of candidate *ZmNAC9*, which plays an important role in response to low P and regulation in root architecture. A significant difference was found under low- and sufficient-P conditions for each of the 22 seedling traits, and a total of 41 polymorphic sites including 32 single nucleotide polymorphisms (SNPs) and 9 insertion and deletions (InDels) were detected in *ZmNAC9* among 111 inbred lines. Among the 41 polymorphic studied sites, a total of 39 polymorphic sites were associated with 20 traits except for the dry weight of shoots and forks, of which six sites were highly significantly associated with a diverse number of low-P tolerant root trait index values by using a mixed linear model (MLM) at −log10 P = 3.61. In addition, 29 polymorphic sites under P-sufficient and 32 polymorphic sites under P-deficient conditions were significantly associated with a diverse number of seedling traits, of which five polymorphic sites (position S327, S513, S514, S520, and S827) were strongly significantly associated with multiple seedling traits under low-P and normal-P conditions. Among highly significant sites, most of the sites were associated with root traits under low-P, normal-P, and low-P trait index values. Linkage disequilibrium (LD) was strong at (r2 > 1.0) in 111 inbred lines. Furthermore, the effect of five significant sites was verified for haplotypes in 111 lines and the favorable allele S520 showed a positive effect on the dry weight of roots under the low-P condition. Furthermore, the expression pattern confirmed that *ZmNAC9* was highly induced by low P in the roots of the P-tolerant 178 inbred line. Moreover, the subcellular localization of *ZmNAC9* encoded by protein was located in the cytoplasm and nucleus. Haplotypes carrying more favorable alleles exhibited superior effects on phenotypic variation and could be helpful in developing molecular markers in maize molecular breeding programs. Taken together, the finding of this study might lead to further functions of *ZmNAC9* and genes that might be involved in responses to low-P stress in maize.

## 1. Introduction

Maize (*Zea mays* L.) is one of the most important food and feed crops globally [1,2]. Improving maize yield is the most important goal of maize breeding programs [3]. It is also obvious that plants suffer from various abiotic stresses throughout their life cycles such as drought, heat, cold, salinity, and nutrient deficiency [4,5,6,7]. Phosphorus (P) is an essential macronutrient for plant growth and development and cell functions cannot be performed by any alternative nutrient [8,9]. Phosphorus has a certain association with water regulation; i.e., osmosis, photosynthesis regulations, plant metabolic activity, and drought resistance [10,11]. A phosphate (Pi) deficiency not only affects plant growth, development, and crop yield but also deteriorates the quality of the fruit and seed formation [12]. However, improving plant tolerance to P deficiency is an efficient alternative source to increase crop production. Therefore, clarifying P-deficient responses in plants at the molecular level is vital for developing improved genotypes that might perform well under low-P conditions. 

However, plants take up P from soil through root hairs to cope with low-P stress and have evolved diverse adaptive mechanisms at physiological, morphological, and molecular levels to enhance P utilization [13,14]. In previous studies, *Arabidopsis thaliana* increased their root hairs by inhibiting the growth of primary roots and several lateral roots in order to adapt under low-P conditions [13,15,16,17]. Generally, plants increase the number of root hairs when exposed to low-P stress [8,17]. The adaptation mechanism of the acid phosphatase gene *AtPAP26* mutant was controlled due to low P [18]. In earlier studies, many acid phosphatase genes were cloned from maize, barley, rice, and wheat where these genes were induced to be expressed under low-P stress [19]. Recently, many researchers have identified the molecular mechanism of plants under low-P stress conditions which contained the suppressor of yeast gpa1, the yeast phosphatase 81, and the human Xenotropic and Polytropic Retrovirus receptor 1 (SPX) domain encoding proteins to phosphate and plants adapting low-P conditions [20,21]. However, many genes containing SPX domains to maintain P in homeostasis in plants play an important role in P signaling and dynamic balancing based on the structure and these SPX domain proteins are divided into four subfamilies including SPX, SPX-RING, SPX-MFS, and SPX-EXS [22]. The five major family members of phosphorus transporters are PHO1, PHO2, PHT1, PHT2, and PHT3, of which the PHT1 family is mainly responsible for P absorption and phosphorus transport [23]. However, the PHO1 protein family is only one protein family containing both SPX and EXS domains [22]. In the previous study, the AtPHO1 mutant gene was identified in wild and mutant varieties of *Arabidopsis* and phosphorus content increased in mutant roots as compared to wild and decreased in the leaves while shoot growth was affected and inhibited in mutants [24]. Later, the AtPHO1 was cloned in *Arabidopsis* containing both the SPX and EXS domain which was found to be involved in the loading process of inorganic P in the xylem of *Arabidopsis* roots [25]. However, two regulatory genes PHR1 and PHR2 are affected under P-deficient conditions. PHR1, micro399, and PHO2 respond to the low-P stress signal pathway.

Association analysis is a method to identify the genetic variation among the natural populations to establish genetic markers based on linkage disequilibrium (LD) [26]. LD or allelic association refers to a group of non-random associations between different loci alleles in the population [27]. Genetically, R2 and D’ are used to estimate the degree of LD between two alleles [28]. Candidate gene association studies the kinship and population structure mostly used to detect the false-positive association [29]. This model is useful for predicting the potential of different populations for an association analysis and response to marker-assisted selection. The association between non-random alleles of markers reflects the size of chromosomal segments remaining in a complete population [30,31]. Maize is a cross-pollinated crop, which is prone to genetic recombination, gene mutation, etc. LD declined rapidly during the process of evolution, so the use of candidate gene association analysis in maize can identify effective alleles [32,33]. In recent years, genome wide association study (GWAS) analysis is the most widely used method to identify candidate genes and resolves some complex quantitative trait variations. In previous studies, 36 loci were significantly associated with 32 small-scale diseases of leaf in the maize population [34,35]. Five candidate genes were identified under low-P stress conditions, as were genes involved in the metabolic pathway and that are responsive to low-P tolerant traits at the seedling stage [36]. The single nucleotide polymorphism (SNP) sites, insertion and deletions (InDels), and SSR markers were developed from candidate genes and involved in phenotypic variation [37,38,39]. In previous studies, some functional molecular markers were associated with specific traits in maize [38]. Two functional molecular markers, HO06 and DGAT04, were associated with high oil content in maize [39]. 

The NAC (NAM, ATAF and CUC) transcription factors plays an important role in developmental and environmental stress-responsive mechanisms and transcriptional regulatory networks associated with plant stress [40,41,42]. The NAC acronym originates from three different protein (NAM, ATAF and CUC) domains [41,42]. Recently, many members of the NAC family have been identified by the availability of complete genome sequence in plants against biotic and abiotic stresses—for example, 117 NAC genes were identified in *Arabidopsis*, 151 genes in rice, 79 genes in grape, 26 genes in citrus, 163 genes in poplar, and 152 genes in soybean, tobacco, and also in other species [22,43,44,45]. In maize, 124 homologous members of the NAC family have been identified based on whole-genome duplication which may lead to an evaluation, molecular functions, and identification analysis in maize [44]. Three novels NAC candidate genes, *ZmNAC145*, *ZmNAC51*, *ZmNAC72*, were induced by drought resistance in maize [45]. Moreover, *ZmNAC84* responded against drought stress in maize [46]. In total, 148 NAC (*ZmNAC1*–*ZmNAC148*) non-redundant genes were identified in a genome-wide association study in maize and all genes belonging to the d group and play a role in plant growth [47]. NAC transcription factors (TFs) contained different domains with at least 150 amino acids in the N-terminal region and these are responsible for DNA binding in the nucleus while the C-terminal region is related to transcriptional regulation and diverse functions [48]. In rice, Os04g0477300 (encoded NAC protein) was identified which is responsible for multiple processes including pathogenic diseases, B toxicity tolerance, boron and P deficiency [49,50,51,52]. 

However, the findings of previous studies provided useful information about NAC transcription factors, such as the fact that NAC genes can regulate diverse functions and may be involved in several processes to improve nutrient efficiency and might be mediated with different signal pathways. These studies have shown the importance of natural variation in genes during the evolution of stress resistance and can be used directly for genetic improvement in maize. In the previous study, a total of 108 candidate genes were identified in a genome-wide association study (GWAS) in maize [36], and *ZmNAC9* was a reliable candidate gene that was not only associated to root dry weight with the low-phosphorus tolerant trait index, but also a differential expression gene in the transcriptome data under different phosphorus condition. Therefore, in this study we re-sequenced *ZmNAC9* in 111 maize inbred lines and aimed to investigate the association between the novel *ZmNAC9* candidate gene with low phosphorus stress-related seedling traits under normal and low-P stress conditions. Besides, molecular markers were identified in maize inbred lines, which can be used as a reference for exploring the molecular mechanism of the *ZmNAC9* gene. Furthermore, the expression pattern was investigated in low-P tolerant 178 and P-sensitive 9782 inbred lines under low-P stress conditions and provides new insights to improve the low-P tolerance in maize and the findings of this study will be helpful to understand the molecular mechanism of candidate *ZmNAC9* under low-P stress conditions in maize.

## 2. Results

### 2.1. Basic Molecular Information of ZmNAC’s Gene Family

A total of 208 ZmNAC family genes were identified in maize. Homologous genes of *ZmNAC9* were identified using a protein sequence by a protein blast tool search similarity in NCBI, maize GDB (https://www.maizegdb.org/) and Gramene online databases at 1 × 10^10^ value. The results showed that the *ZmNAC9* consisted of 780 bp and encoded putative protein 259 amino acids (aas), with an isoelectric point (pI) of 9.70, and six homologous genes were similar to *ZmNAC9* based on high similarity scores. The CDS of *ZmNAC9* homologous genes were between 594 and 1053 bp and their proteins were between 197 and 350 amino acids with one, two or three transcripts. The molecular weight was different among genes analyzed by ExPASy (http://www.expasy.org/) online software and the isoelectric point varied between pI 5.52 and pI 10.64 (Table 1). The subcellular localization of genes was predicted by the cello online database (http://cello.life,netu.edu.tw) and most of the genes were found to be located in the nucleus [51].

### 2.2. Phylogenetic Tree from Different Species 

To identify the evolutionary relationship of *ZmNAC9* and their homologous genes within maize, rice and *Arabidopsis thaliana*, a tree was constructed by using MEGA 7.0 with a neighbor-joining method by the amino acid sequence of *ZmNAC9* homologous genes from rice and *Arabidopsis thaliana* and the uniform rate was tested using 1000 bootstrap replicates (Figure 1). Results showed that *ZmNAC9* and their homologous genes were in closed relationships with rice members, respectively.

### 2.3. Gene Structure, Domain and Chromosomal Location of ZmNAC9 and Their Homologous Genes in Maize

To detect the evolutionary relationship of the ZmNAC gene family, six ZmNAC homologous genes of *ZmNAC9* were selected from 10 subclasses of an evolutionary tree. Among the 10 subclasses, six genes belonged to the same subclade. Results showed that these genes contained 3 exons and 2 introns and conserved the domain which might provide support to the relationship of genes homologous with *ZmNAC9* (Figure 2). Besides, we performed chromosomal locations with a physical relative distance model. Results showed that *ZmNAC9s* and their homologous genes were distributed on different chromosome numbers and two homologous genes were on seven chromosomes in maize (Figure 3).

### 2.4. Sequence Polymorphism and Nucleotide Diversity of ZmNAC9 Gene

The total genomic DNA reference sequence of the candidate ZmNAC9 (1459 bp) gene was downloaded from the maize GDB (https://www.maizegdb.org/) database. The candidate *ZmNAC9* gene was successfully amplified in 111 maize inbred lines. The coding sequence region of the *ZmNAC9* gene was 780 bp and a total of 32 single nucleotide polymorphisms (SNPs) and 9 insertion and deletions (InDels) were identified with a minor allele frequency (MAF) >0.05 in the coding region of *ZmNAC9* among 111 inbred lines—the results are shown in Appendix A. The results revealed that the average nucleotide diversity of *ZmNAC9* was Pi (π) 0.01055 for all inbred lines, for polymorphic sites this was 35, for haplotypes 16 and haplotypes diversity was 0.861 in the coding region of *ZmNAC9* among 111 maize inbred lines. Besides, nucleotide diversity was analyzed using a sliding window of 100 bp with a 25 bp step size, and two parameters—π and θ values—were used to measure nucleotides diversity. The results indicated that the π value was rapidly decreased at 0.005 around 200 bp and slightly increased at 0.03 and maintained around 800 bp. While the θ value decreased at 0.005 and slightly increased at 0.01 and minted around 400 bp, it then increased at 0.02 and maintained around 800 bp (Figure 4). Additionally, the Tajima D test was mostly applied for neutral theory in the genetic population to measure the allelic frequency in the nucleotide sequence. In the present study, the Tajima D test exhibited non-significance at *p* < 0.05 whereas Fu and Li’s tests showed a significant neutral variation in the coding region of *ZmNAC9* at *p* < 0.05 in the tested maize population (Table 2). Even though these results cannot deny the hypothetical mutation drift equilibrium, a lack of a footprint of positive selection in *ZmNAC9* was suggested. It also implies that there are functional components of the gene-encoding proteins in the process of evolution.

### 2.5. Linkage Disequilibrium (LD) Analysis in the Coding Sequence of ZmNAC9 Gene

LD is an effective method to investigate the functional allelic variation in association mapping analysis. The LD was analyzed using Tassel software v5.0. In this experiment, the LD was analyzed between all pairs of polymorphic sites (SNPs) and insertion deletion polymorphisms (InDels) in the coding region of the ZmNAC9 gene in 111 inbred lines. The significant LD was found at (r2 > 0.1) and also various polymorphic sites at (*p* < 0.00001) in 111 maize inbred lines and different polymorphic sites showed a different level of LD (Figure 5A), respectively. Besides, the LD decay of ZmNAC9 was rapidly decreased in 500 bp physical distance base pairs at (r2 > 0.1) (Figure 5B), respectively.

### 2.6. Association Analysis of Low-P Tolerant Traits with ZmNAC9 Gene

The phenotypic difference of 22 seedlings traits was calculated in three independent treatments under low-P, normal-P, and low-P tolerant trait index values. Therefore, 22 significant traits were evaluated for association with the ZmNAC9 gene. A total of 41 polymorphic loci were identified in the coding region of the ZmNAC9 gene including 32 SNPs and 9 InDels. The association between 41 polymorphic sites with 22 low-P tolerance traits was performed using a statistically standard mixed linear model (MLM+Q+K) including a population structure and kinship matrix and a threshold was set at (–log P value 3.61) –log10(0.01/41 polymorphic loci). Among 41 polymorphic sites, 39 sites were associated with 20 traits and a total of 191 associations were significant with the P-tolerant traits index except for the dry weight of shoot and forks. Of which six sites (S327, S513, S514, S520, S827, and S839) were highly significantly associated with a diverse number of P-tolerant trait index values (Figure 6a, Table 3 and Appendix A). At the same time, among 41 sites 29 sites were associated with 22 traits, of which six polymorphic sites (position S327, S513, S514, S520, S633, and S839) were highly significantly associated with a variable number of low-P seedling traits under normal-P conditions (Figure 6b, Table 3 and Appendix A). However, among 41 polymorphic loci, 32 polymorphic loci included SNPs and InDels were significantly associated with the different number of seedling traits and the remaining nine polymorphic loci were non-significantly associated with each trait under low-P conditions (Figure 6c, Table 3 and Appendix A). Of them five sites (position S242, S513, S521, S824, and S827) were strong significantly associated with multiple seedling traits using the same threshold under low-P stress conditions. Among all significant sites, most of the sites were associated with root traits with low-P, normal-P and low-P trait index values. Linkage disequilibrium was strong at (r2 > 1.0) in 111 inbred lines. 

### 2.7. Haplotypes Diversity in the Coding Sequence of ZmNAC9 Gene

Haplotypes were identified in the coding sequence of the *ZmNAC9* gene. A total of 15 haplotypes were identified in tested 111 maize inbred lines in the *ZmNAC9* gene. Among 15 haplotypes, haplotype-1 contained 25 inbred lines of which the most frequent was CDS haplotype 1 and other haplotypes contained more than 10 inbred lines included hap-2, hap-3, hap-4 comprising 23, 13, and 12 inbred lines, respectively (Table 4 and Appendix A). Besides, the effects of polymorphic loci on the dry weight of low-P tolerant trait index values were evaluated by haplotype analysis in ZmNAC9. The results showed that, among the identified haplotypes, five major haplotypes contained more than 10 inbred lines which emerged from five significant sites and these haplotypes were used for a pairwise haplotype comparison analysis. hap4 with a more favorable allele showed the highest significant difference with hap2, hap3, and hap5 while was non-significant with hap1 at *p* < 0.05. A pairwise comparison analysis and one-way ANOVA test were applied to determine significant differences between haplotypes (Figure 7). 

### 2.8. Expression Pattern of ZmNAC9 Gene in Low-P Tolerant and P-Sensitive Inbred Lines by qRT-PCR

A gene expression analysis is important in many fields of biological research. Understanding the expression pattern of genes provides useful information about complex regulatory networks occurring in living organisms and gene functions. In this study, the expression pattern of *ZmNAC9* was examined in two maize inbred lines, P-tolerant 178 and P-sensitive 9782, under low-P stress treatments during the seedling growth stage by using quantitative real-time polymerase chain reaction (qRT-PCR). Results showed that the expression pattern was increased in the early stage but decreased in the later stage of P treatment, and the *ZmNAC9* gene was up-regulated on 1, 3, 7 and 9 days in roots and 1, 3, 7, 9, and 12 days in the leaves of the P-tolerant 178 inbred line under low-P treatments, while down-regulated at others time points. In the P-sensitive 9782 inbred line, the *ZmNAC9* gene was induced on 1, 3, 12 and 16 days in roots and 1, 3, 9 days in leaves (Figure 8). The expression pattern was different between 178 and 9782 under low-P stress treatments and results suggested that the *ZmNAC9* gene was more induced in P-tolerant 178 under P-deficient conditions. 

### 2.9. Subcellular Localization of ZmNAC9 Gene 

Transcription factors (TFs) are typically localized in the nucleus of the cell and perform a transcriptional activation role and DNA binding. Therefore, in the present study the subcellular localization of *ZmNAC9* was detected in tobacco (Nicotiana benthamiana) leaves by using the transient transformation method and tested method by green fluorescence protein (GFP) and the fusion protein of *ZmNAC9*-eGFP driven by the 35S promoter. The GFP fluorescence signals of *ZmNAC9*-eGFP were localized in the nucleus and cytoplasm by using laser confocal microscopy (LSCM). The empty vector pCAMBIA2300-35S-eGFP without any foreign gene fragment was observed in the nucleus (Figure 9). These results were not consistent with those predicted by the CELLO v.2.5 program [53], which expected transcription factors to be only located in the nucleus. Generally, model plant tobacco or onion epidermal cells were mostly used for subcellular localization due to successful results. However, the subcellular location of genes is the main issue in cell biology and important study in molecular biology [54]. 

## 3. Discussion

Maize is one of the model plants used for basic and advanced studies of plant domestication and genetic improvement due to its genetic diversity and large phenotypic variation [55,56]. Phosphorus is an essential major element for plant growth and yield and induces biological processes (e.g., when there is a P deficiency) that affect the development, growth, yield, and also the biological functions of plants [10,33,57]. NAC proteins play vital roles in plant development, including signaling pathways, and act as regulators in various abiotic and biotic stresses [44,47]. In this study, we identified the 208 ZmNACs family members in maize and these family members were classified into 10 subclades on a phylogenetic tree. Among 10 subgroups the *ZmNAC9* and their homologous genes were on five subclades according to their position. Besides, the homologous genes were BLASTP with Arabidopsis thaliana and rice, and the phylogenetic tree was constructed in maize, Arabidopsis thaliana, and rice, indicating the evolutionary process of homologous genes in diverse species. Although, the homologous genes of *ZmNAC9* were distributed in different subgroups with members of Arabidopsis and rice, this indicated that the evolutionary process of these genes had not let to functional difference but it might provide a reference to find the functions of maize the same as previously reported for the NAC family in maize [47]. Furthermore, homologous genes were evaluated for the distribution of chromosomal locations, gene structure, and conserved motif. The homologous genes were distributed at different chromosomal locations. The chromosome location indicated that two homologous genes of *ZmNAC9* were distributed on seven chromosomes at a short distance. The gene structure of *ZmNAC9* homologous genes showed that all genes had a similar number of exons and conserved motif and results indicated that all genes have the same functions, such as stress responses. Our findings are in agreement with previous studies that the NAC family has a highly conserved domain [48,58]. Therefore, *ZmNAC9* might have the same functions as their homologous genes in maize as well as in other species.

The natural variation mostly caused by genetic variation is in the ancestor of maize teosinte [59]. Previous studies showed that the average nucleotide frequency of one SNP included 104 bp on 1 chromosome [60]. One single nucleotide polymorphism per 31 bp was found in non-coding regions and one polymorphism per 124 bp was found in the coding regions of 18 maize genes in 36 elite maize inbred lines [61]. In this study, candidate gene association was performed on 111 maize inbred lines at the seedling stage under low-P and normal-P conditions. A total of 41 polymorphic loci including 32 single nucleotide polymorphism (SNP) and 9 insertion and deletion (InDels) markers were identified in the coding sequence region of *ZmNAC9* in 111 inbred lines with a minor allele frequency of *p* < 0.05 (Table 3 and Appendix A). Less nucleotide diversity was found in the *ZmNAC9* gene with one polymorphic site. Similar results were previously observed, finding that the level of nucleotide diversity might be contributed to by gene properties in different population sizes and diverse plant species and the nucleotide might have occurred on an average of 47.7, 61 and 104 bp in the gene region [62,63,64]. The findings of this study suggested that the SNPs and InDels could be essential to generate new changes during the evolution of maize species. Besides, Tajima’s D and Fu and Li’s D* tests showed that *ZmNAC9* had non-significant neutral variation in the coding region based on Tajima’s D and significant variation on Fu and Li’s D* tests at *p* < 0.10. This was a not-obvious artificial selection and the negative value of genes indicated that the neutral selection hypothesis could not be excluded and the gene was naturally selected in the screening of germplasm. The effect of selection is small but it is easier to produce gene drift and so on. The results were changed only in the sequence of genes in maize inbred lines. The findings of this study are similar with previous studies that found that the artificial selection of genes might affect the coding region of genes and depends upon the number of germplasm [57,62,63]. Furthermore, LD analysis was analyzed because it is an essential method to examine the genomic variation for the association mapping of complex traits [64]. In our study, the LD level was high in some sites and low in other sites which might be due to genetic mutations in the gene segment. Besides, the LD decay of *ZmNAC9* declined lower than the expected r2 value within 500 bp (Figure 5A). LD decay indicated that *ZmNAC9* was less affected by natural variation and it was not under the selection pressure during germplasm and these loci might be functional sites for the response to low phosphorus (Figure 5B). Our results are in agreement with the previous studies that LD decay could be different between 200 and 2000 bp and selective sweep in maize lines [52]. 

Moreover, the candidate gene association can identify the natural variation and is best on haplotypes for targeted trait responses to low-P stress conditions in maize at the seedling stage. In this study, we used 22 traits for the *ZmNAC9* association analysis. The results revealed that *ZmNAC9* has good potential in low-P stress; among 41 polymorphic sites, 39 sites were associated with 20 traits and a total of 191 associations were significant with the P-tolerant traits index except for the dry weight of shoots and forks using the mixed linear model (MLM) including Q+K. Six sites were highly significantly associated with a diverse number of P tolerant trait index values (Figure 6a, Table 3 and Appendix A). At the same time, among 41 sites 29 sites were associated with 22 traits, of which six polymorphic sites (position S327, S513, S514, S520, S633, and S839) were highly significantly associated with the variable number of low-P seedling traits under normal-P conditions (Figure 6b, Table 3 and Appendix A), whereas, 32 polymorphic loci included SNPs and InDels were significantly associated with the different number of seedling traits, and the remaining nine polymorphic loci were non-significantly associated with each trait under low-P conditions (Figure 6c, Table 3 and Appendix A). Five sites (position S242, S513, S521, S824, and S827) were strong significantly associated with multiple seedling traits using the same threshold under low-P stress conditions. Furthermore, haplotypes were analyzed in the tested population. Results showed that among the identified haplotypes five major haplotypes contained more than 10 inbred lines, of which hap4 with a more favorable allele showed the highest significant difference at *p* < 0.05. Previous studies have identified favorable natural variation in candidate genes—for example, *ZmDREB2.7*, *ZmNAC111*, *ZmPP2C-A*, *ZmVPP1*, and *ZmHKT1* (5, by using the MLM statistical model) [13,57,65,66]. 

Plants have evolved many physiological, biochemical, morphological, and molecular mechanism adaptation strategies to cope with P-deficiency stress [13]. Recently, many researchers have been identified the phosphorus transporter members in different crops such as, PHO1, PHO2, PHT1, PHT2, and PHT3 five major family members, of which most member of PHT1 family mainly responsible for phosphorus absorption from the soil, distribution within plants and up-regulated under low-P stress condition [23]. In our study, the expression pattern was increased in the early stage but decreased in the later stage of low-P treatment and *ZmNAC9* was dominant in roots but also increased in leaves and was significantly induced by the low-P stress condition in the P-tolerant 178 and P-sensitive 9782 inbred lines. There is no report about the P transporter *ZmNAC9* in maize. Results suggested that ZmNAC9 was probably involved in response to the P-starvation condition and might play an important role in maize and improve the low-P tolerance in maize. 

The functions and characterization of any gene product in a post-genome sequence was the main challenges for plant biologists because it can be limiting to understand the biological roles of many genes [65]. The localization of gene expression products in cells is the main issue in cell biology and an important study in molecular biology [54]. Therefore, we identified the subcellular localization of *ZmNAC9* through the transient transformation method and a targeted fragment was fused with the green fluorescence protein (GFP) introduced in tobacco leaves driven by the CamV35S promoter and the green fluorescence protein (GFP) of *ZmNAC9* was detected in nucleus and cytoplasm (Figure 9). This is different from the natural transcription factors only localized in the nucleus by the predication cello program [51]. The subcellular localization of *ZmNAC9* might be disturbed due to various cellular processes because NAC transcription factors have a different conserved (NAM, ATAF and CUC) domain and each domain has a diverse function and environmental stress and the location might be changed [66].

## 4. Materials and Methods

### 4.1. Plant Materials, Growth Conditions and Phenotyping

A total of 111 diverse genotypically maize inbred lines (46 tropical 65 temperate), were screened under low-P stress conditions to identify the low-P tolerant inbred lines from the Southwest China breeding program [67]. These diverse heterotic groups of 111 maize inbred lines were used for a candidate gene association study (Appendix A). The phenotypic traits difference was carefully identified after 15 days of low-P-stress and control P-sufficient condition. The manually evaluated traits were: primary root length (PRL); root maximum length (MRL); plant height (PH); number of seminal roots (NSRs); number of crown roots (NCRs); number of leaves (NLs); shoot fresh weight (FSW); seminal root fresh weight (FWSR); primary root fresh weight (FWPR); crown root fresh weight (FWCR); total root fresh weight (FWR); total plant fresh weight (FWP); root biomass (BR); shoot biomass (SB); root/shoot ratio (R/S); total plant biomass (DWP). The maize root architecture system was classified according to the protocol described in [68]. In addition, the total root length (TRL), root surface area (RS), root average diameter (AD), root volume (RV), number of root tips (T), and number of root forks (F) under low- and normal-P conditions have been previously analyzed [36]. These twenty-two significant traits were used for natural variation in the *ZmNAC9* gene.

### 4.2. Bioinformatics Analysis of NAC TFs Family Members in Maize

The amino acid sequence of *ZmNAC9* gene was used as a query to search for the exact number of ZmNAC candidate genes in the maize family by using the Pfam http://pfam.sanger.ac.uk and http://http://smart.embl-heidelberg.de/ online databases. To evaluate the evolutionary relationship of ZmNAC family members in maize, the protein sequences of ZmNAC members were aligned in ClustalW to generate the phylogenetic tree of the ZmNAC family members of maize by using the neighbor-joining method in MEGA v7.0 [69]. The uniform rate was tested using 1000 bootstrap replicates and named according to their position correction in the phylogenetic tree. Besides, the relationship of *ZmNAC9* and its homologous genes was compared with other species, and the protein sequence of *ZmNAC9* and its homologous genes were used in the BLASTP program against Arabidopsis and rice with an E-value of 1 ×10^−10^. The phylogenetic tree was constructed using the same methods as ZmNACs family members. Furthermore, the basic molecular characteristics of *ZmNAC9* and its homologous genes were identified through the ExPASy online database (https://www.expasy.org/) and genes were summarized according to their physical and chemical properties such as the accession number of genes, genes on a chromosome, number of transcriptions, molecular weight, isoelectric points, and grand average of hydropathicity (GRAVY) for each putative protein. Furthermore, the chromosome location of *ZmNAC9* homologous genes was downloaded from the Gramene database and chromosomal location distance was calculated by using Map-Draw software 2.1 as previously described [70]. In addition, the gene structure of *ZmNAC9* homologous genes including the exons, introns, the 5’ and 3’ UTR regions were evaluated through the gene structure display server (GSDS) online program [71]. Moreover, the prediction of the conserved domain of ZmNAC9 homologous genes was evaluated through the MEME online prediction database using the full length of an amino acid’s sequence of genes. The parameter settings are as follows: the maximum number of motifs was 5 and length of the amino acid was 6–600 (aa). 

### 4.3. Sequencing Polymorphism, Nucleotide Diversity and LD of ZmNAC9 Candidate Gene

The total genomic DNA of 111 inbred lines was extracted from fresh roots and leaves by using the cetyltrimethyl ammonium bromide (CTAB) method described in [72]. The genomic DNA sequence of *ZmNAC9* of the B73 inbred line reference sequence was downloaded from Maize GDB (http://www.maizegdb.org). The forward (AGTGTTTGGACGGGGTTGAGTCCATTTCA) and reverse (AGCGTTGCTGGAAGGGAACTGATTGATTT) primers were designed according to the target sequence of a gene by using primer premier 5v software. The high fidelity phanta max enzyme polymerase was used for the amplification of a targeted fragment of genes and DNA was used as a template; the total reaction volume was 25µL, containing 12.5 μL phanta max buffer, 0.5 μL dNTP, 0.5 μL phanta max super fidelity, 8.5 μL ddH2O, 1 μL genomic DNA and 1 μL of each sense and anti-sense. The PCR cycle procedure was set as: initial denaturation at 95 °C for 3 min followed by 30 cycles of denaturation at 95°C for 15 sec and an annealing step for 15 s at 63°C, extension at 72°C for 1.15 min, an extension for 5 min at 72 and 12 °C for preservation. Amplified fragments for all inbred lines were determined by gel electrophoresis (Biorad, California) using 1% agarose gels in Tris-EDTA (TE) buffer. Agarose gel was stained with 2 uL of ethidium bromide and the running time was 25 min at 220 mV and then target fragments were direct and subjected to TsingKe Biological Technology Co., Ltd for sequencing. The targeted sequence of the ZmNAC9 gene was compared with the reference genomic sequence of the B73 inbred line by using DNAMAN v6.0 software. The sequence was adjusted manually by using Bio-Edit 7.1 software [56], and a multiple sequence alignment of ZmNAC9 gene was performed for 111 inbred lines by using clustalX 2.0 [73]. We analyzed single nucleotide polymorphisms (SNPs) and insertion and deletions (InDels) in all tested lines with a <0.05 minor allele frequency by using Tassel software v5.0 [74]. Furthermore, the linkage disequilibrium (LD) between two polymorphic sites in the coding region of the ZmNAC9 gene was generated in an LD plot by using TASSELv5.0 [74]. In addition, the average of the r2 values was used to calculate the LD decay with physical distance and a loess regression test was applied by using the R language in the gg plot package. In addition, nucleotide diversity, nucleotide polymorphism, haplotype diversity, haplotypes, haplotype distribution, and Tajima’s D and Fu and Li’s D tests were analyzed among 111 sequenced maize inbred lines for a neutrality test by using DnaSp v5.0 [75]. Furthermore, a sliding window of 100 bp with a 25 bp step size and π and θ values were used to measure the nucleotides’ diversity.

### 4.4. Candidate ZmNAC9 Gene Association Analysis

An association analysis between single nucleotide polymorphism (SNPs) and insertion and deletion (InDel) sites in the *ZmNAC9* gene with 22 phenotypic traits were calculated using Tassel software v5.0 [74]. The standard mixed linear model (MLM) included a population structure and kinship matrix which were considered for false-positive control in association analysis under low-P stress conditions as previously described in [55]. In which both the population (Q) and kinship (k) matrix were calculated as earlier described [67]. The identified 32 SNPs and 9 InDels polymorphic loci with a MAF at *p* ≥ 0.05 were used for candidate gene association analysis. The association analysis was considered significant at *p* < 0.01 and the calculated p-values were converted into −log10(P) to improve the accuracy of the association between polymorphic loci and low-P tolerant traits under low-P, normal-P conditions and the corresponding (T/CK) low-P tolerant trait index using a uniform Bonferroni-corrected test as described [36,76].

### 4.5. Low-P Stress Treatments, RNA Extraction and Expression Pattern of Candidate ZmNAC9 Gene

The seeds of 178 P-tolerant and 9782 P-sensitive inbred lines were provided by Maize Research Institute, Sichuan Agricultural University, Chengdu, China. Seeds were soaked at 37 °C for 3 days. After soaking, seeds were surface-sterilized using a 10% sodium hypochlorite solution for 20–30 min. Seeds were rinsed with distilled water two to four times and kept for 12 h at 28 °C and then transferred into a plastic vessel (5 L capacity of distill water) fitted with brown thermocol sheets. Standardized Hoagland solution was applied in plastic vessels for the initial three days of seedling to adapt to the hydroponic environment. The composition of Hoagland nutrients contained low P (−P, 1 μmol l−1 KH_2_PO_4_) and sufficient P (+P, 1 mmol l−1 KH_2_PO_4_). The pH was maintained for a nutrient solution around 5.5 and the aquarium pump was used during the experiment. Root and leaves were harvested separately at 0 h, 1, 3, 7, 9, 12, 14, and 16 days under sufficient-P and deficient-P conditions. Total RNA was extracted from each sample according to the manufacturer’s protocol using TRIzol Reagent® (Life Technologies now ThermoFisher Scientific, https://www.thermofisher.com). Reverse transcription of cDNA using a Prime Script TM II 1st Strand cDNA Synthesis Kit and cDNA was used as a template using Fast Start Essential DNA Green Master (Roche, https://www.roche.com) on a Roche Cobas Z480 system. The specific primers were designed using BEACON DESIGNER 7 software; qRT-PCR was programmed in initial step at 95 °C for 10 m, followed by 40 cycles including denaturation at 95 °C for 10 s, annealing at 58.4 °C for 30 s and extension at 60 °C for 15 s. The myosin housekeeping gene was used as an internal control and data were calculated in the 2^−ΔΔCT^ method as described [77] with at least three biological repeats for each sample and three technical repeats were performed for each biological replicate. Three technical replications were used to calculate the average and standard deviation (SD) of expression levels for each sample.

### 4.6. Subcellular Localization of ZmNAC9 Gene

The full-length CDS region of the *ZmNAC9* gene was cloned using the specific primers; forward ATTTGGAGAGGACAGGGTACCATGGAGGCCCGGCCCGG (SAM1 underlined enzyme site) and reverse GGTACTAGTGTCGACTCTAGAATTAGAGGAGGCGACGCTGCAGCT (Xba1 underlined enzyme site) were added in both primers. The subcellular localization was assayed in tobacco leaves according to the reported transit transformation method [78,79]. The fragment of the gene was fused with pCAMBIA2300-35S-GFP; the position of *ZmNAC9* gene was between the cauliflower mosaic virus (CaMV) 35S promoter and GFP gene. The empty pCAMBIA2300 vector without GFP was used as a control and the loaded vector pCAMBIA2300-35S; *ZmNAC9*-eGFP (recombinant plasmid) was transformed into the *Agrobacterium tumefaciens* EHA105 strain and then introduced into tobacco (*Nicotiana benthamiana*) leaves as previously described [80]. The transformed tobacco epidermal cells were observed by the A1R-si laser confocal microscope (LSCM, Nikon, Japan) to imagine the GFP fluorescent signals. At least three times the experiment was repeated for consistent results.

## 5. Conclusions

In this study, a total of 208 ZmNAC family members were identified and *ZmNAC9* homologous genes, their molecular characteristics, gene structure, and conserved domains were studied. A bioinformatics analysis did not lead the functional difference but provided the reference for the functional analysis. Furthermore, the natural variation of *ZmNAC9* revealed an association of candidate gene with low-P tolerant traits and might allow the investigation of molecular markers under low-P stress conditions and genetic background. In addition, we validated the expression pattern of *ZmNAC9* under low-P stress by using the low-P tolerant 178 and P-sensitive 9782 inbred lines. Our results provide useful information for the improvement of low-P tolerance in maize and to understand the molecular characteristics and functions of *ZmNAC9* under low-P conditions in future molecular breeding programs in maize.

## Figures and Tables

**Figure 1 plants-09-01447-f001:**
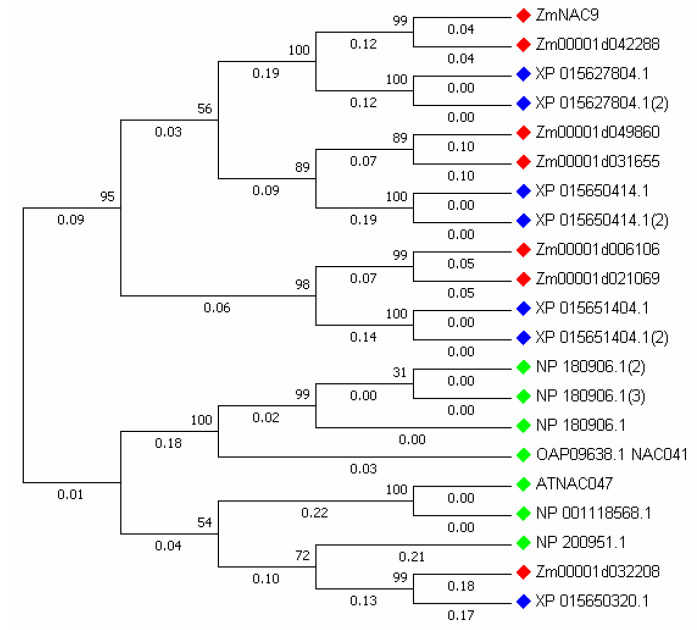
The phylogenetic tree of *ZmNAC9* and their homologous genes. Red mark indicates the homologous *ZmNAC9*; blue—rice genes and green—*Arabidopsis thaliana*.

**Figure 2 plants-09-01447-f002:**
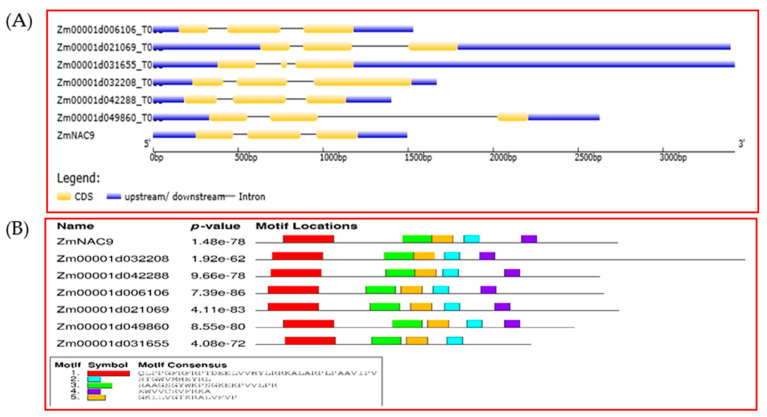
The gene structure and conserved domain of maize 7 genes. (**A**) Untranslated and exons are represented in blue and yellow boxes while introns are presented by blank line. (**B**) Conserved domains are presented in different color boxes.

**Figure 3 plants-09-01447-f003:**
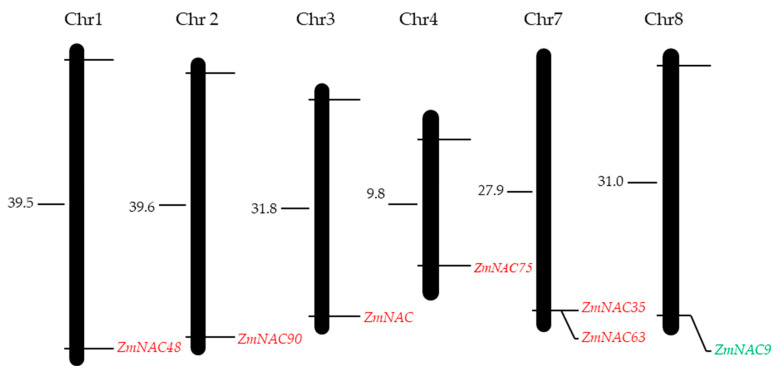
Chromosomal distribution of *ZmNAC9* and their homologous genes. *ZmNAC9* indicated by green color while homologous by red color. Numbers on chromosomes represented by relative physical distance. Chr—chromosomes.

**Figure 4 plants-09-01447-f004:**
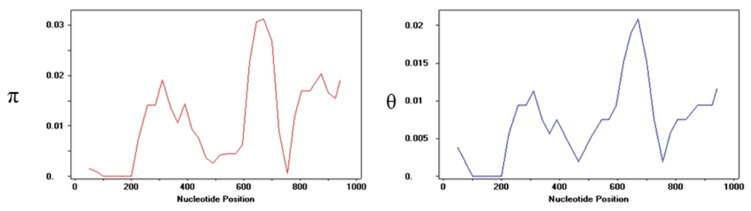
The nucleotide diversity π and θ value in the coding region of *ZmNAC9* using the sliding windows of 100 bp with the step of 25 bp in 111 inbred lines.

**Figure 5 plants-09-01447-f005:**
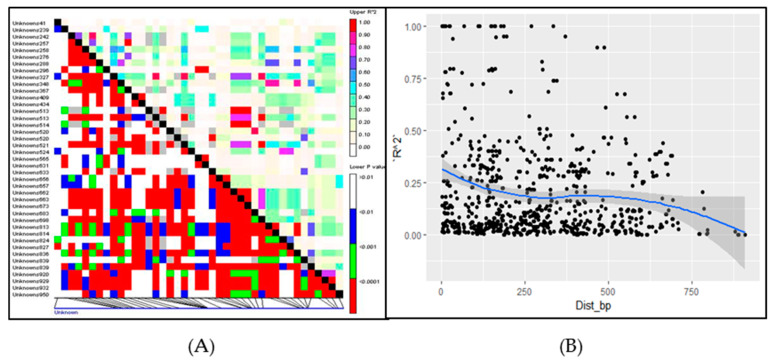
(**A**) Linkage disequilibrium pattern; (**B**) linkage disequilibrium (LD) decay of *ZmNAC9* in 111 maize lines.

**Figure 6 plants-09-01447-f006:**
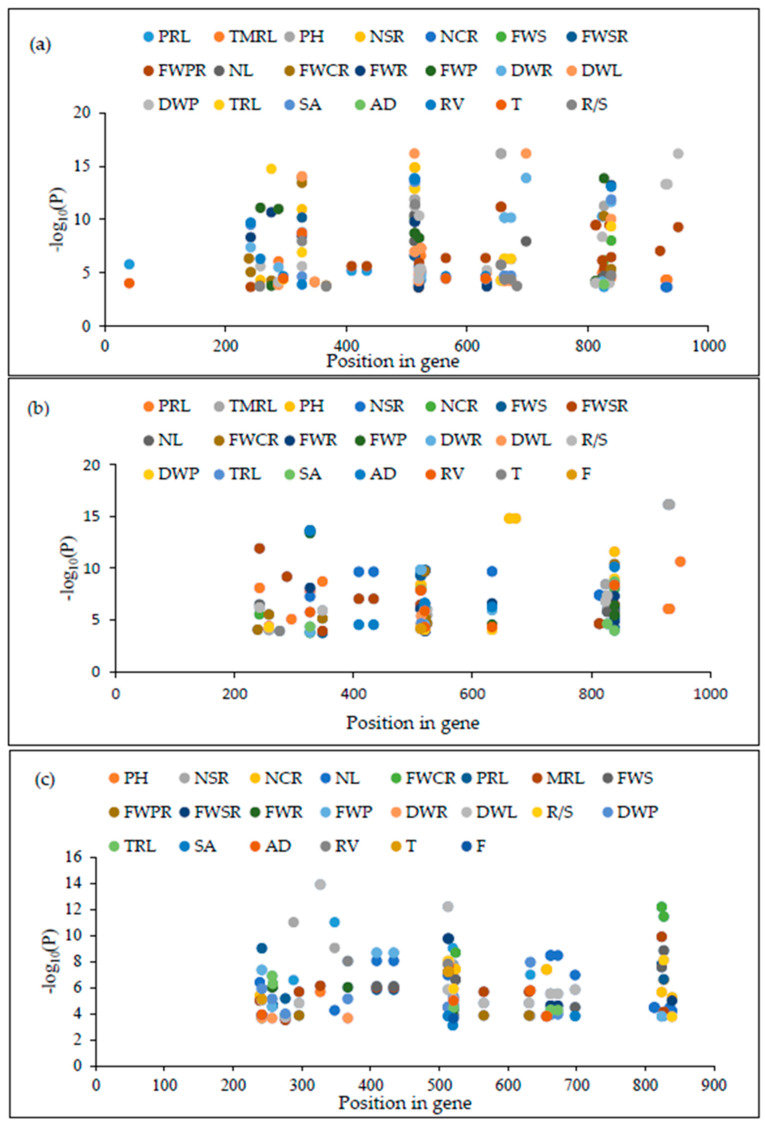
Association analysis of polymorphic sites of *ZmNAC19* and low-phosphorus tolerance seedling traits at threshold (–log P value 3.43). (**a**) Polymorphic sites with low-phosphorus (P) tolerant trait index; (**b**) polymorphic sites with low-P tolerant trait under normal-P conditions; (**c**) polymorphic sites with low-P tolerant trait under low-P conditions.

**Figure 7 plants-09-01447-f007:**
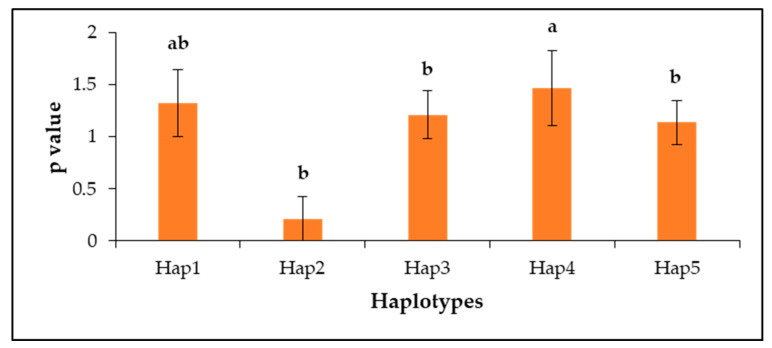
Effect of haplotypes consisting of five significant loci on low-phosphorus tolerance index of dry weight of roots in 111 inbred lines and a, b and ab represented the significance difference.

**Figure 8 plants-09-01447-f008:**
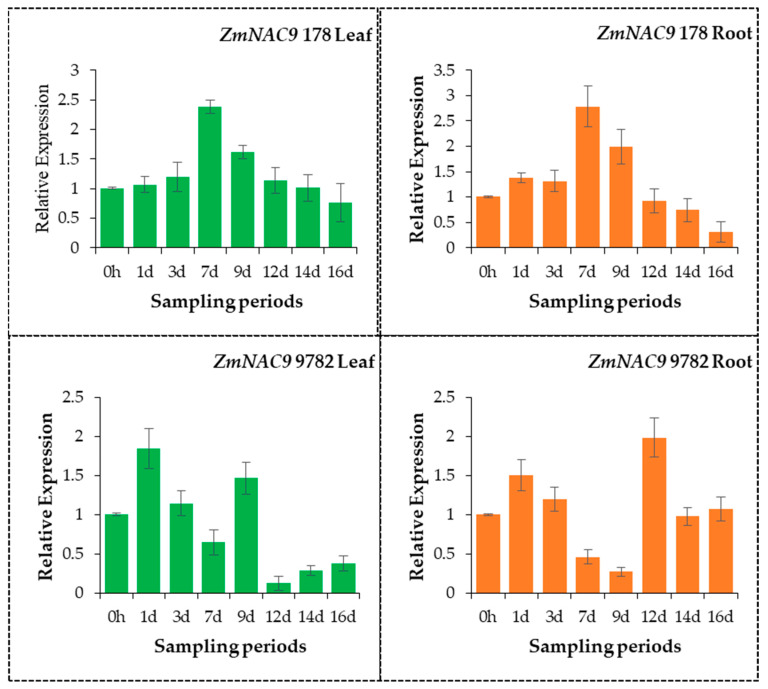
Expression pattern of ZmNAC9 in roots and leaves of P-tolerant 178 and P-sensitive 9782 inbred lines under low-P conditions in maize seedling.

**Figure 9 plants-09-01447-f009:**
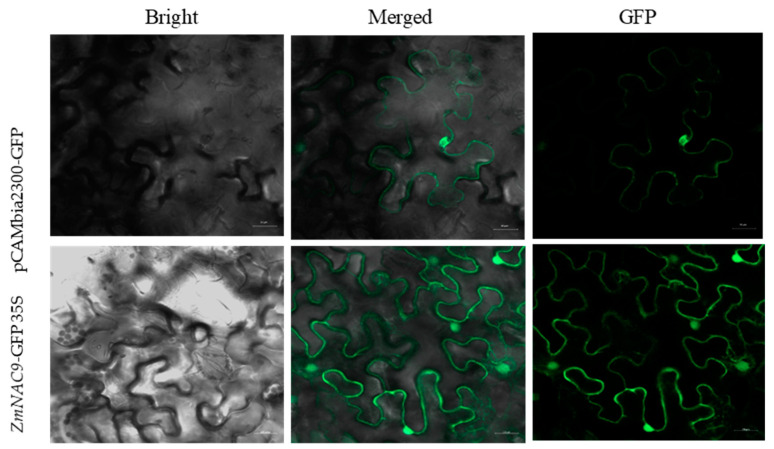
Subcellular localization of *ZmNAC9*–green fluorescence protein (GFP) green fusion protein in the tobacco (Nicotiana benthamiana) epidermal cells of leaf.

**Table 1 plants-09-01447-t001:** Molecular characteristics of *ZmNAC19* homologous genes in maize.

Gene Names	Accession Number	CDSbp	Amino Acids	Ch	T	kDa	pI	GRAVY	Subcellular Location
*ZmNAC9*	Zm00001d011969	780	259	8	2	25,390.81	9.70	−0.327	N
*ZmNAC*	Zm00001d032208	1053	350	3	1	39,270.58	5.70	−0.680	N
*ZmNAC90*	Zm00001d042288	741	246	2	3	26,239.40	8.21	−0.510	N
*ZmNAC35*	Zm00001d006106	783	260	7	1	28,013.65	10.07	−0.582	N
*ZmNAC63*	Zm00001d021069	750	249	7	2	27,645.22	10.07	−0.752	N
*ZmNAC75*	Zm00001d049860	687	228	4	1	24,313.91	10.46	−0.288	N
*ZmNAC48*	Zm00001d031655	594	197	1	3	21,592.24	10.64	−0.178	M/C/P

Note. Ch, T, kDa, pI, GRAVY, N, C, M, C and P stand for chromosome, transcripts, molecular weight, isoelectric point, grand average of hydropathicity, nucleus, cytoplasm and mitochondrial plasma membrane, respectively.

**Table 2 plants-09-01447-t002:** Summary of single nucleotide polymorphism (SNP) sites, nucleotide diversity (π), nucleotide polymorphism (θ) and Tajima’s D, Fu and Li’s D test of *ZmNAC9* gene in a different maize population, and ** representing the significance value, *p* < 0.05.

TF	Coding Region	SNP/InDels	Nucleotide Diversity	Nucleotide Polymorphic Sites	Haplotypes	Haplotypes Diversity (Hd)	Tajima’s D	Fu and Li’s D *
*ZmNAC9*	780 bp	32/10	Pi(π)0.01055	35	16	0.861	1.47198	1.73019 **

**Table 3 plants-09-01447-t003:** Statistics of association analysis between the polymorphisms in *ZmNAC9* and low-P tolerance traits in maize seedlings.

Site No.	Loci Name	Position in Gene (bp)	SNPs	Minor Allele Frequency	MLM(Q+K) Number of Significant Traits			
					T/CK Index	T	CK	–log10 *P* Value
1	S41	41	C/G	0.05405	3	0	0	ns
2	S239	239	C	0.17117	1	3	1	*
3	S242	242	C	0.10909	6	10	5	***
4	S257	257	C/T	0.10811	2	5	1	**
5	S258	258	G/A	0.3964	4	3	5	**
6	S276	276	T	0.14414	4	5	1	***
7	S288	288	A/T	0.47748	5	1	1	**
8	S296	296	T/A	0.09009	4	3	1	**
9	S327	327	C	0.40541	13	3	10	***
10	S348	348	G/A	0.45946	2	2	5	**
11	S367	367	G/C	0.10811	2	4	0	*
12	S409	409	G/A	0.24324	2	5	3	**
13	S434	434	T/A	0.24324	2	5	3	**
14	S513	513	G/A	0.26415	16	11	10	***
15	S513	513	G	0.26415	0	0	0	ns
16	S514	514	C/A	0.26415	14	5	11	***
17	S520	520	C/T	0.25773	11	5	18	***
18	S520	520	0/3	0.10834	0	0	0	ns
19	S521	521	G/C	0.28866	6	10	5	***
20	S524	524	G/C	0.11712	7	3	3	**
21	S565	565	C/T	0.09009	4	3	0	**
22	S631	631	T/C	0.09009	4	3	0	**
23	S633	633	A/T	0.12613	5	3	8	***
24	S656	656	T/G	0.37838	5	2	0	**
25	S657	657	G/T	0.37838	5	2	0	**
26	S662	662	A/G	0.2973	6	5	1	***
27	S663	663	T/C	0.2973	6	5	1	***
28	S673	673	G	0.2973	6	5	1	***
29	S683	683	C/A	0.23423	1	0	0	ns
30	S698	698	T/G	0.25225	3	4	0	**
31	S813	813	A/G	0.48649	3	1	2	**
32	S814	814	G/C	0.48649	3	1	2	**
33	S824	824	A/G	0.0991	6	6	2	***
34	S827	827	A	0.12037	9	7	5	***
35	S836	836	G/A	0.48649	3	1	2	**
36	S839	839	G/A	0.28261	11	4	20	***
37	S839	839	A	0.28261	1	0	0	ns
38	S920	920	G/A	0.21622	3	0	0	ns
39	S929	929	T/C	0.26126	3	0	2	**
40	S932	932	C/T	0.26126	3	0	2	***
41	S950	950	T/C	0.27027	2	0	1	**

Note: The letters represent nucleotides; letters represent the major alleles and underlined letters minor alleles. T/CK indicate low-P tolerant trait index; T represents P-deficient conditions; CK, normal-P conditions; MLM (Q+K), mixed linear model, population structure and kinship matrix; numerical numbers indicate the significant traits and – indicates data unavailable, ** indicates the –log10(0.01/41), *** indicate the –log10(0.001/41) according to the previously described method for correction test for p-values [52].

**Table 4 plants-09-01447-t004:** Division of five haplotypes and corresponding base changes in ZmNAC9 gene.

Haplotypes	Lines	S327	S513	S514	S520	S524	S827
**Hap1**	25	C	T	G	C	G	0 bp
**Hap2**	23	C	T	G	T	G	0 bp
**Hap3**	13	C	A	G	C	C	0 bp
**Hap4**	12	C	T	G	C	G	0 bp
**Hap5**	10	C	T	G	C	C	16

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
