# Peer review of "Identification, Association of Natural Variation and Expression Analysis of *ZmNAC9* Gene Response to Low Phosphorus in Maize Seedling Stage"

_plants, 2020, doi:10.3390/plants9111447_

Round 1

Author Response

Response Letter

Dear editor,

 We are very grateful and thankful to you and reviewers for considering our manuscript. Please consider the revised version of our research article. Identification, Association of Natural variation and Expression Analysis of ZmNAC9 Gene Response to Low Phosphorus in Maize Seedling Stage “to be published in the journal of “Plants”. In the following pages are our point-to point responses to each of the comments raised by the reviewers. We hope that the revisions in the manuscript and our accompanying responses will be sufficient to make the manuscript suitable for publication in Plants. We believe that our study will be interesting for the readers of Plants as well as broad community of plant scientists.

Your sincerely,

Javed Hussain Sahito

Sichuan Agricultural University

Comments and Suggestions for Authors

Reviewer 1:

Reviewer #1:

The manuscript investigated the association between novel ZmNAC9 candidate gene with low phosphorus stress-related seedling traits under normal and low P stress conditions. Authors present interesting big data. The concluded that natural variation of ZmNAC9 revealed an association of candidate gene with low P tolerant traits validating its expression pattern. Even though, the manuscript assessed an interesting topic, the English was really hard to read. In general, among the several changes requested the phosphorus word is used after it was abbreviated before as P, you should decide if you will use phosphorus or P. Also, there are many words written in capital letter when they should be written in lower case. The term of low P stress, means a low stress of P or stress due to low P?? The manuscript must be rewritten in order to improve the understanding of the study.

Response:

Thank you so much, for suggestions. In order to improve the flow and understanding of text for readers. We have carefully checked and made substantial improvements in the revised version. Redundant sentences have been deleted, miss-leading statements also have been corrected from the whole text and revised text is highlighted in red. Moreover, we have substantially improved the grammar and quality of english writing as per your suggestion.

2: Line 22 Please correct “limiting factor of maize growth and yield”

Response:

Thank you very much for your careful reading. Please note the following changes in the revised version. Line number 22.

3: Line 23 Please add abbreviation of phosphorus as (P)

Response:

Thank you for your comment and suggestion, we have highlighted in revised manuscript Line number 22.

4: Line 28 Please correct “Among of the 41 polymorphic studied sites”

Response:

We have modified and corrected in revised manuscript. Line number 28

5: Line 29 What’s the meaning of Pi, please add first the meaning with the corresponding abbreviation

Response:

Thank you for your valuable comments, Pi means inorganic phosphorus because Phosphorus (P), an indispensable macronutrient for plant growth and development, is absorbed by root systems in the form of Pi from soil. We have modified and highlighted in revise manuscript. Line number 32 to 36

6: Line 32 Please correct Moreover, from the 41 studied sites, 29 sites were” also, please change the word Whereas is not correct to use it on that context

Response:

We have modified in revised manuscript. Line number 33 to 35

7: Line 38 and 41 Is confusing, please correct the phrase

Response:

We have modified in revised manuscript. Line number 37 to 41

8: Line 48 staple food?

Response:

We have highlighted in revised manuscript. Line number 49

9: Line 53 Please correct and its functions??

Response:

We have modified in revised manuscript. Line number 53 to 55

Line 54 Please correct the reference ZHANG by Zhang

Response:

Thank you, we have applied MDPI reference in revised manuscript. Line number 52

Line 54 Please use the abbreviation of P

Response:

Thank you for your comment and suggestion, we have used abbreviation of P in revised manuscript. Line number 52

Line 57 Pi was not defined before, please add the meaning of Pi first

Response:

Thank you for your comment and suggestion, we have defined in revised manuscript. Line number 55

Line 57 Please correct the phrase

 Response:

We have highlighted in revised manuscript. Line number 55 to 57

Line 59 What´s the meaning of that? Great potential in soil?

Response:

Thank you for your comment and suggestion, we have highlighted in revised manuscript. Line number 57 to 59

Line 60 When you mention phosphate rock you should explain that P fertilizers are originated from it

Line 61 Please correct phosphorus by P

Line 62 total crop production? Is referring to world production

Response:

Thank you for your suggestion, we have modified in revised manuscript. Line number 58 to 59

Line 63 please correct Plants by plant

Thank you for your suggestion, we have modified in revised manuscript. Line number 60

Line 64 Please correct “P is taken by plants from the soil through root hairs”. Using low P environmental stress is not correct, its mean that is a low stress?? I think you referred to stress given by a low P concentration in soils

Response:

Thank you for your suggestion, we have modified in revised manuscript. Line number 60 to 63

Line 67 Is confusing, please correct the phrase

Response:

Thank you for your suggestion, we have modified in revised manuscript. Line number 63 to 65

Line 70 All in all, should be replace by another word

Line 72 “found” is not well used

Line 78 meaning of SPX structures

Response:

Thank you for your comments and suggestion, we have modified in revised manuscript. Line number 65 to 75

Line 82 Confuse please rewrite

Response:

Thank you for your comments and suggestion, we have modified in revised manuscript. Line number 65 to 75

Line 88 “showed” is not well used

Line 93 Please remove “re”

Response:

Thank you for your comments and suggestion, we have removed in revised manuscript. Line number 78 to 85

Line 95 You already defined LD please use the abbreviation after

Line 97 Please remove the coma

Response:

Thank you for your comments and suggestion, we have modified in revised manuscript. Line number 78 to 85

Reviewer 2 Report

In this manuscript, the authors did in-depth genetic research of the ZmNAC9 gene. They identified the variants that are associated with 22 low phosphorus tolerance traits, found the haplotype and favorable alleles that have positive effects, and observed the ZmNAC9 induction by low Pi in the roots.

Major scientific problems:

  1. What is the genetic background of the 11 inbred lines? Tropical maize and temperate maize have very different low phosphorus response. If the genetic background were very diverse, probably the diversity and the Tajama’D should be calculated for different populations. Is the frequency of desired haplotype are the same in different genetic background ?
  2. In Figure 1, the phylogeny has no branch length. Please add it back. And due to the recently whole-genome duplication, most genes have two copies in the maize genome. Zm00001d042288 is the paralog of ZmNAC9. What is the similarity between Zm00001d042288 and ZmNAC9? For all the candidate ZmNAC9 that amplified in 111 maize inbred lines, they should compare with Zm00001d042288 too to eliminate the probability that you amplified Zm00001d042288 instead of ZmNAC9.

There are several minor problems might help you further improve your manuscript:

  1. The authors identified a total of 41 polymorphic loci in the coding region of the ZmNAC9 gene including (32 SNPs and 9 InDel). Would you provide variants annotation to indicate the effect of these variants, like synonymous mutation, in-frame InDel, or frame-shift InDel?
  2. In figure 8, the limits of the y-axis are different in four panels. If the author makes the limit are the same in the four panels, it will make it clear that the ZmNAC9 is induced more in the P-tolerant 178.  
  3. Figure 7, what is the y-axis?

The biggest issue is the English writing, many sentences are hard to understand and there are some fragmented sentences, for example:

L145: “Although ZmNAC9 was identified in a GWAS study 145 under low phosphorus stress conditions in maize seedling (Luo et al., 2019). “ This is not a complete sentence.

L252: “Table 2. Summary of single nucleotide polymorphism (SNP) sites, nucleotide diversity (p), nucleotide 253 polymorphism ("), and Tajima’s D, Fu and Li's D test of ZmNAC9 gene in a different maize population” what is in that empty (“”)?

Author Response

Response Letter

Dear editor,

 We are very grateful and thankful to you and reviewers for considering our manuscript. Please consider the revised version of our research article. Identification, Association of Natural variation and Expression Analysis of ZmNAC9 Gene Response to Low Phosphorus in Maize Seedling Stage “to be published in the journal of “Plants”. In the following pages are our point-to point responses to each of the comments raised by the reviewers. We hope that the revisions in the manuscript and our accompanying responses will be sufficient to make the manuscript suitable for publication in Plants. We believe that our study will be interesting for the readers of Plants as well as broad community of plant scientists.

Your sincerely,

Javed Hussain Sahito

Sichuan Agricultural University

Comments and Suggestions for Authors

Reviewer 2:

Reviewer #2:

In this manuscript, the authors did in-depth genetic research of the ZmNAC9 gene. They identified the variants that are associated with 22 low phosphorus tolerance traits, found the haplotype and favorable alleles that have positive effects, and observed the ZmNAC9 induction by low Pi in the roots.

Response:

Thank you so much, for suggestions. In order to improve the flow and understanding of text for readers. We have carefully checked and made substantial improvements in the revised version. Redundant sentences have been deleted, miss-leading statements also have been corrected from the whole text and revised text is highlighted in red. Moreover, we have substantially improved the grammar and quality of english writing as per your suggestion.

  1. What is the genetic background of the 11 inbred lines? Tropical maize and temperate maize have very different low phosphorus response. If the genetic background were very diverse, probably the diversity and the Tajama’D should be calculated for different populations. Is the frequency of desired haplotype are the same in different genetic background?

Response:

Thank you for your comment and suggestion. Yes, these 111 inbred lines include tropical and temperate background and the low-phosphorus response are predicted different. We can add the diversity and the Tajama’D calculated for each subpopulation. We also add the frequency of haplotype for different background.

  1. In Figure 1, the phylogeny has no branch length. Please add it back. And due to the recently whole-genome duplication, most genes have two copies in the maize genome. Zm00001d042288 is the paralog of ZmNAC9. What is the similarity between Zm00001d042288 and ZmNAC9? For all the candidate ZmNAC9 that amplified in 111 maize inbred lines, they should compare with Zm00001d042288 too to eliminate the probability that you amplified Zm00001d042288 instead of ZmNAC9.

Response:

Thank you for your valuable comments and suggestions, we have added branch length in the phylogeny. Moreover, we have reconfirmed the sequence of ZmNAC9 amplified in 111 inbred lines to eliminate the ambiguity and find that we did not amplify Zm00001d042288 instead of ZmNAC9.

  1. The authors identified a total of 41 polymorphic loci in the coding region of the ZmNAC9 gene including (32 SNPs and 9 InDel). Would you provide variants annotation to indicate the effect of these variants, like synonymous mutation, in-frame InDel, or frame-shift InDel?

Response:

Thank you for your careful reading, we have followed our previous studies to check the polymorphic site effects. Line number 340 to 347

SAHITO J H, ZHENG F, TANG H, et al. 2020. Identification, association, and expression analysis of ZmNAC134 gene response to phosphorus deficiency tolerance traits in maize at seedling stage. Euphytica [J], 216: 100.

2: In figure 8, the limits of the y-axis are different in four panels. If the author makes the limit are the same in the four panels, it will make it clear that the ZmNAC9 is induced more in the P-tolerant 178.  

Response:

We have modified and corrected in revised manuscript. Figure 8 and Line number 367 to 373

3: Figure 7, what is the y-axis?

Response:

We have mentioned in revised manuscript. Figure number 7

3: L145: “Although ZmNAC9 was identified in a GWAS study 145 under low phosphorus stress conditions in maize seedling (Luo et al., 2019). “ This is not a complete sentence.

Response:

Thank you for your valuable comments and suggestion, we have improved incomplete sentences and mentioned in revised manuscript. Line number 127 to 131

4: L252: “Table 2. Summary of single nucleotide polymorphism (SNP) sites, nucleotide diversity (p), nucleotide 253 polymorphism ("), and Tajima’s D, Fu and Li's D test of ZmNAC9 gene in a different maize population” what is in that empty (“”)?

Response:

We have modified in revised manuscript. Line number 33 to 35

Round 2

Reviewer 1 Report

The manuscript was improved according to the previous suggestions, while authors must decide to use the abbreviation of phosphorus as P or write it as phosphorus.

Line 60, 61, 65, 66, 68,69, 71, 72, 73, 75, 76 and others. Please use the abbreviation of P

Line 69 Add: where these genes were induced to be expressed under low P stress.

Line 70 Please correct researchers instead of researcher

Line 72 Please explain the meaning of SPX before

Author Response

Response Letter

Dear editor ,

 We are very grateful and thankful to you and reviewers for considering our manuscript. Please consider the revised version of our research article. Identification, Association of Natural variation and Expression Analysis of ZmNAC9 Gene Response to Low Phosphorus in Maize Seedling Stage “to be published in the journal of “Plants”. In the following pages are our point-to point responses to each of the comments raised by the reviewers. We hope that the revisions in the manuscript and our accompanying responses will be sufficient to make the manuscript suitable for publication in Plants. We believe that our study will be interesting for the readers of Plants as well as broad community of plant scientists.

Your sincerely,

Javed Hussain Sahito

Sichuan Agricultural University

Comments and Suggestions for Authors

Reviewer 1:

Reviewer #1:

The manuscript was improved according to the previous suggestions, while authors must decide to use the abbreviation of phosphorus as P or write it as phosphorus.

Response:

Thank you very much, for you valuable suggestions. We have carefully checked in the revised manuscript version and highlighted in Track Changes" function in Microsoft Word. Moreover, we have substantially improved as per your suggestion.

2: Line 60, 61, 65, 66, 68,69, 71, 72, 73, 75, 76 and others. Please use the abbreviation of P

Response:

Thank you for careful reading, we have used abbreviation of P in revised manuscript. Line number 60, 61, 65, 66, 68, 69, 71, 72, 73, 75, 76 and others

Line 69 Add: where these genes were induced to be expressed under low P stress.

Response:

We have added in our revised manuscript. Line number 69

Line 70 Please correct researchers instead of researcher

Response:

We have corrected in our revised manuscript. Line number 70

Line 72 Please explain the meaning of SPX before

Response:

Thank you very much for your comments, SPX means Suppressor of Yeast gpa1 (Syg1), the yeast Phosphatase 81 (Pho81), and the human Xenotropic and Polytrophic Retrovirus receptor 1.  We have mentioned in revised manuscript. Line number 72 and the detailed information of SPX is in following manuscript

Du, H., Yang, C., Ding, G., Shi, L., & Xu, F. (2017). Genome-wide identification and characterization of SPX domain-containing members and their responses to phosphate deficiency in Brassica napus. Frontiers in plant science, 8, 35
